# Global and regional burden estimation of HIV-associated non-Hodgkin's lymphoma: a meta-analysis and modelling analysis protocol

Yan Chen ,[1,2] Zhaochen Sun,[1,2] Ping Sun,[2] Yuping Liu,[2] Zhengwei Wan,[2] Yunli Ye[1]

¹School of Public Health, Southwest Medical University, Luzhou, Sichuan Province, China
²Department of Health Management Center & Institute of Health Management, University of Electronic Science and Technology of China, Chengdu, Sichuan Province, China

**Correspondence to**
Professor Yunli Ye;
wushuangyewu@163.com

## ABSTRACT

**Introduction** HIV infection is one of the complex aetiologies of non-Hodgkin's lymphoma (NHL). However, the contribution of HIV to burden of NHL across time and region has not yet been comprehensively reported and quantified. Thus, this study aims to evaluate the relative risk of NHL in individuals with HIV infection compared with those without by performing a comprehensive meta-analysis. Additionally, we intend to further estimate quantitatively the degree of HIV contributing to burden of NHL using population attributable fraction (PAF) modelling analysis.

**Methods and analysis** This study will screen a mass of records searched from four electronic databases (PubMed, Embase, Cochrane Library and Web of Science). The main outcomes are specific effect values and corresponding 95% CIs for NHL among population with HIV infection compared with those without to quantify the association between HIV infection and NHL. After quality assessment and data extraction, we will undertake a meta-analysis to calculate the pooled risk ratio (RR). Furthermore, PAF calculation based on pooled RR combines with number of age-specific disability-adjusted life year (DALY) and HIV prevalence data (aged ≥15 years old) from 1990 to 2019, at global, regional and country levels. We will calculate the PAF, HIV-associated DALY number and age-standardised rate to quantify the burden of HIV-associated NHL.

**Ethics and dissemination** This study is based on published articles; thus, the ethic approval is not essential. In addition, we intend to publish the results on peer-reviewed journals for more discussion. We believe that research on estimating global burden of NHL can provide valuable insights for developing targeted prevention and control strategies, thereby achieving significant benefits.

**PROSPERO registration number** CRD 42023404150.

## STRENGTHS AND LIMITATIONS OF THIS STUDY

⇒ The study will include meta-analysis and population attributable fraction (PAF) estimation modelling analysis, which provides aid in the formulation of precise prevention and control policies.
⇒ The quality of eligible studies will be assessed using an 11-item checklist recommended by the Agency for Healthcare Research and Quality.
⇒ The dissemination of the results may be limited because of the language restriction.
⇒ The pooled risk ratio value will be applied to PAF modelling across global and regional areas, which may lead to bias.

## INTRODUCTION

Non-Hodgkin's lymphoma (NHL) is a common lymphoid neoplasm that originates from B cell precursors, mature B cells, T cell precursors and mature T cells.[1] According to the WHO classification, the most common NHL in western countries is diffuse large B cell lymphoma (accounting for around 31% of adult cases), follicular lymphoma (22%) and Burkitt lymphoma (2%).[2 3] According to Global Cancer Observatory database of WHO's International Agency for Research on Cancer, NHL accounted for 2.8% of total cancers with 544 352 new cases, and 2.6% of total cancer-related deaths with 259 793 new deaths, ranking NHL within the top 10 cancers for both males (ranked 8th) and females (ranked 10th) in 2020.[4]

The aetiology of NHL exhibits significant complexity. Published articles have identified various potential causes of NHL, including immunosuppression, viral infection (eg, HIV), family factor, occupational and environmental exposures, organ transplantation and autoimmune diseases.[5–7] NHL is regarded as a defining event of AIDS,[6] thus establishing a strong connection between HIV and NHL. Based on a population-based review encompassing the USA, Australia and Italy, the risk of NHL among individuals with AIDS compared with general population, ranged from 15-fold for low-grade NHL to 400-fold for high-grade NHL.[8] In addition, certain subtypes of NHL have been found to be associated with HIV infection. Another multicohort study conducted in Europe

revealed a higher incidence of NHL among individuals with HIV, regardless of whether they received combination antiretroviral therapy (cART) treatment.[9] Although the introduction of cART in the late 1990s has significantly reduced the incidence of NHL,[10 11] the USA alone recorded 74 680 new NHL cancer diagnoses and 19 910 NHL-related deaths in 2018. Meanwhile, the high burden of HIV infection in South Africa has contributed to an increased incidence of plasmablastic lymphoma.[6 12]

Despite the strong association between HIV infection and NHL, the contribution of HIV to burden of NHL across time and region has not been comprehensively reported and quantified yet. Therefore, this study aims to perform a meta-analysis to quantify the association of HIV with NHL. Additionally, we intend to further estimate the degree of HIV contributing to burden of NHL using population attributable fraction (PAF) modelling analysis. The anticipated outcome of this study is expected to provide valuable insights for the development of targeted prevention and control strategies for NHL.

## Objectives

This study aims to resolve questions:
1. Although HIV infection has a strong connection with NHL, there is no article quantifying this association comprehensively.
2. The contribution of HIV infection to burden of NHL across time and region has not been reported and quantified yet.

## METHODS AND ANALYSIS

Our study will comprise two essential components as follows: (1) meta-analysis to quantify the association between HIV infection and NHL; (2) PAF modelling analysis to assess HIV-associated burden of NHL.

**Table 1** Data to be extracted from eligible articles

| Category | Data |
|---|---|
| Publication details | Author, publication year |
| Type of design | Cohort study, CCS, or cross-sectional study with effective control group |
| Information of study | Study period, NHL subtype, country and region of study conducted, study setting (multicentre or single centre), disease classification criterion, data source or name of cohort registered, median patient age |
| Effect values | RR, OR and SIR and their corresponding 95% CI |
| Study participants | Total number of HIV patients, total number of control group, number of NHL in the case/control group |

CCS, case-control study; CI, confidence interval; HIV, human immunodeficiency virus; NHL, non-Hodgkin's lymphoma; OR, odds ratio; RR, risk ratio; SIR, standardised incidence rate.

## Meta-analysis to quantify the association between HIV infection and NHL

### Registration

This protocol will follow to the guidelines of Preferred Reporting Items for Systematic reviews and Meta-Analyses for Protocols 2015[13] and has registered on the PROSPERO (https://www.crd.york.ac.uk/PROSPERO/) (CRD 42023404150).

### Outcomes definition, inclusion criteria and exclusion criteria

The main outcomes of this review are specific effect values and corresponding 95% CIs for NHL among individuals with HIV infection compared with those without to quantify the association between HIV and NHL. Effect values that will be considered are risk ratio (RR), odds ratio and standardised incidence rate.

For inclusion criteria, our target design types are cohort study, case-control study or cross-sectional study with effective control group. These studies need to expound the association between HIV infection and NHL using target effect value. In addition, eligible studies should provide detailed participant information, such as the total number of participants, total number of HIV patients, total number of control group and number of NHL patients in case/control groups.

For exclusion criteria, we will exclude records whose types are review, opinion pieces, poster presentations, case reports or case series and letters. If there is no targeted effect value or study population, this record will be also excluded. In addition, when there exists overlap population in multiple articles, we will choose the one with larger study scale and higher article quality. As regards duplicate records, we will exclude incomplete or earlier period version.

### Search strategy

We will search for English articles without restrictions of date or setting from four electronic databases: PubMed, Embase, Cochrane Library and Web of Science up to 31 March 2023 with keywords "non-Hodgkin lymphoma", "human immunodeficiency virus", "Burkitt lymphoma", "follicular lymphoma", and "diffuse large B-cell lymphoma" (ie, NHL, subtypes of NHL, HIV and their equivalent terms). In addition, we will also screen references from eligible studies to ensure the integrity of the searches. The full electronic search strategies are available at the additional information (online supplemental appendix 1).

### Data management and study selection

All retrieved records will be imported into EndNote V.X9.2 to manage and screen eligible studies. Research members will enact selection procedures and inclusion criteria in advance all together.

### Step 1: duplicate record and article type screening

Two reviewers will initially use the built-in function of Endnote V.X9.2 to eliminate duplicate records. Subsequently, they will employ keyword searches within the

software to exclude review and case reports. Due to the limitation of incomplete duplication-checking function, certain duplicate records still need to be removed manually. Thus, we leverage the advantage of two-person independent operation to reduce the possibility of omissions. Following the manual deletion process, the two reviewers will cross-check the exclusion results to ensure the highest level of completeness in the final record selection.

### Step 2: title screening
Next, two authors will review the titles of the remaining records, excluding those that do not involve HIV population or outcome of NHL. Moreover, these records that contain mice, mouse, review, case report, drug trial or other non-target exposure and outcome will also be excluded based on our target study type and study subject.

### Step 3: abstract screening
In the abstract, if any record contains the target effect values and accords with desired causal association, it will proceed to the next screening phase. If there is any uncertainty or content deficiency, it is necessary to read the full text.

### Step 4: full-text screening
The full text of remaining records will be downloaded from databases. Two authors will read full text throughout to reconfirm and extract data that can be used for statistical analyses. Articles will be ultimately included if they meet the eligibility criteria. Throughout the screening process, any discrepancy or conflict will be solved by consensus or arbitration from a third reviewer. Additionally, meticulous records will be maintained for each step of the screening processes, which will facilitate the creation of a clear flow chart and the composition of formal article.

### Data extraction
Two authors evaluate independently all retrieved articles by title, abstract and full text according to inclusion and exclusion criteria. Solution of discrepancy or conflict will be the same as the part of study selection. EndNote V.X9.2 will be used to manage the citations and extract data. The following data will be considered to extract (table 1).

### Quality assessment
All included studies will be assessed independently by two authors using an 11-item checklist recommended by the Agency for Healthcare Research and Quality (https://

**South Africa in 2000:**

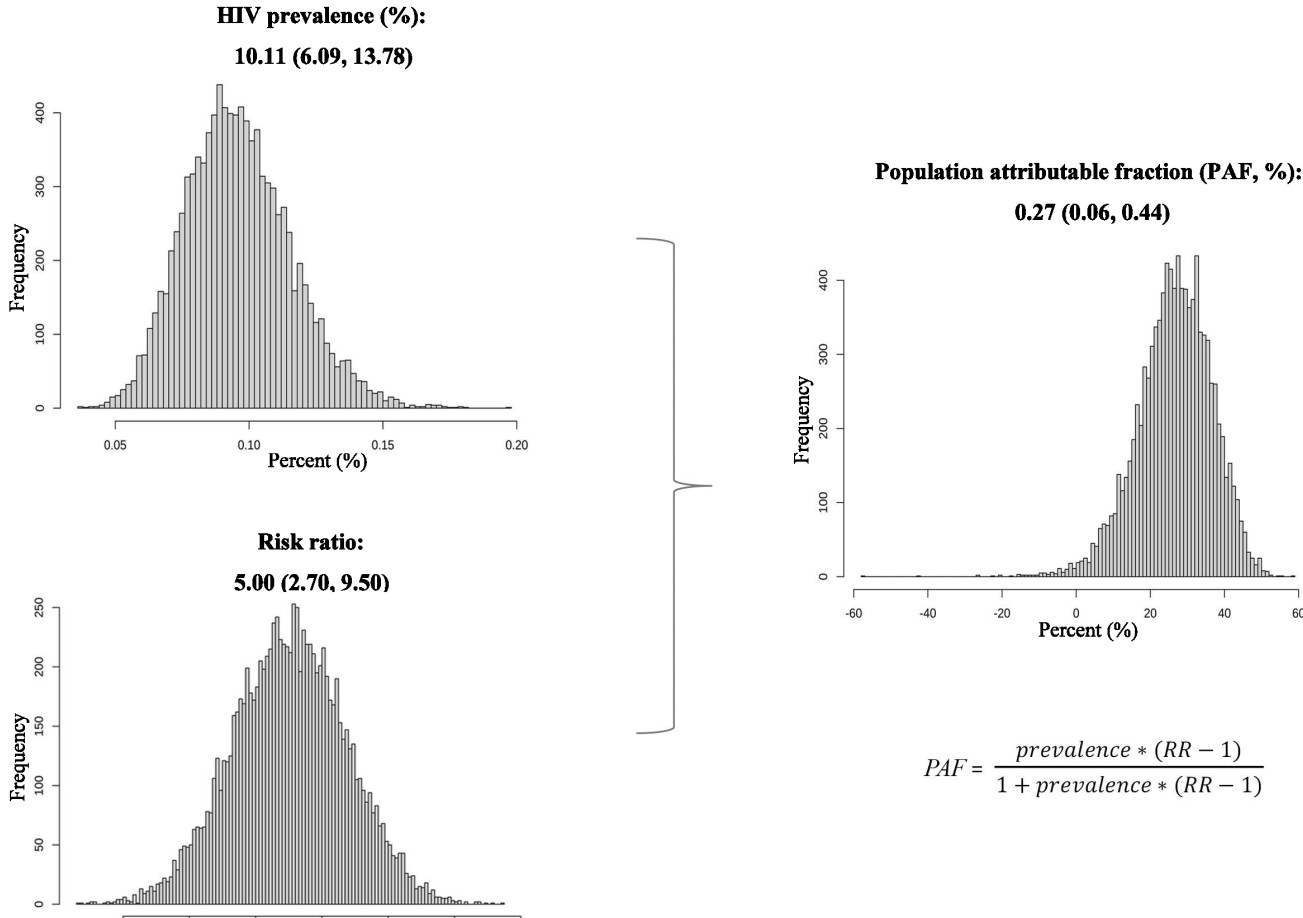

$$PAF = \frac{prevalence * (RR - 1)}{1 + prevalence * (RR - 1)}$$

**Figure 1** Example procedure for calculating PAF in South Africa. PAF, population attributable fraction.

www.wjx.cn/m/85284180.aspx). In this checklist, an item would be scored as '0' if the answer is 'NO' or 'UNCLEAR', and '1' if the answer is 'YES'. The determination of study quality will be complied with correspondence relationships as follows: low quality (0–3), moderate quality (4–7) and high quality (8–11).[14] The judgement of studies' risk level will accord to the result of study quality assessment to divide into high-risk, middle-risk and low-risk levels corresponding to low, middle and high quality in sequence. Any disagreements between two authors will be resolved according to the method outlined in the study selection process.

### Statistical analysis

Overall statistical analyses and calculation will be completed by the R software (V.4.2.2). First, we will calculate a pooled RR and corresponding 95% CI using function 'metagen' included in package 'meta' to quantify the association between HIV infection and NHL. Then, the heterogeneity will be assessed using the $\chi^2$ test on Cochrane's Q statistic, and the quantitation of heterogeneity will be measured by Higgins's $I^2$ value presented as a percentage.[15] When the $I^2$ value exceeds

50%, we will choose random-effect model; otherwise, a fixed-effect model will be used. If we can obtain sufficient data, subgroup analyses will be used to find the potential source of heterogeneity, including NHL subtype, study period, population age, study conducted region, study design, setting of study (multicentre or single centre), and article risk level. Furthermore, a metaregression will be performed to ensure whether these factors can explain the heterogeneity and the degree they account for it. Regarding sensitivity analysis, we intend to analyse influence of single study by omitting one by one. And we will remove studies with different risk levels to judge the robustness of the results. Moreover, we will use Egger's linear regression and funnel plot (if >10 individual studies) to find potential publication bias from these articles. If publication bias exists, we will measure it using the trim-and-fill method.

### PAF modelling analysis for HIV-associated burden of NHL estimation

#### Information sources

The number of age-specific disability-adjusted life year (DALY) will be obtained from the Global Health Data

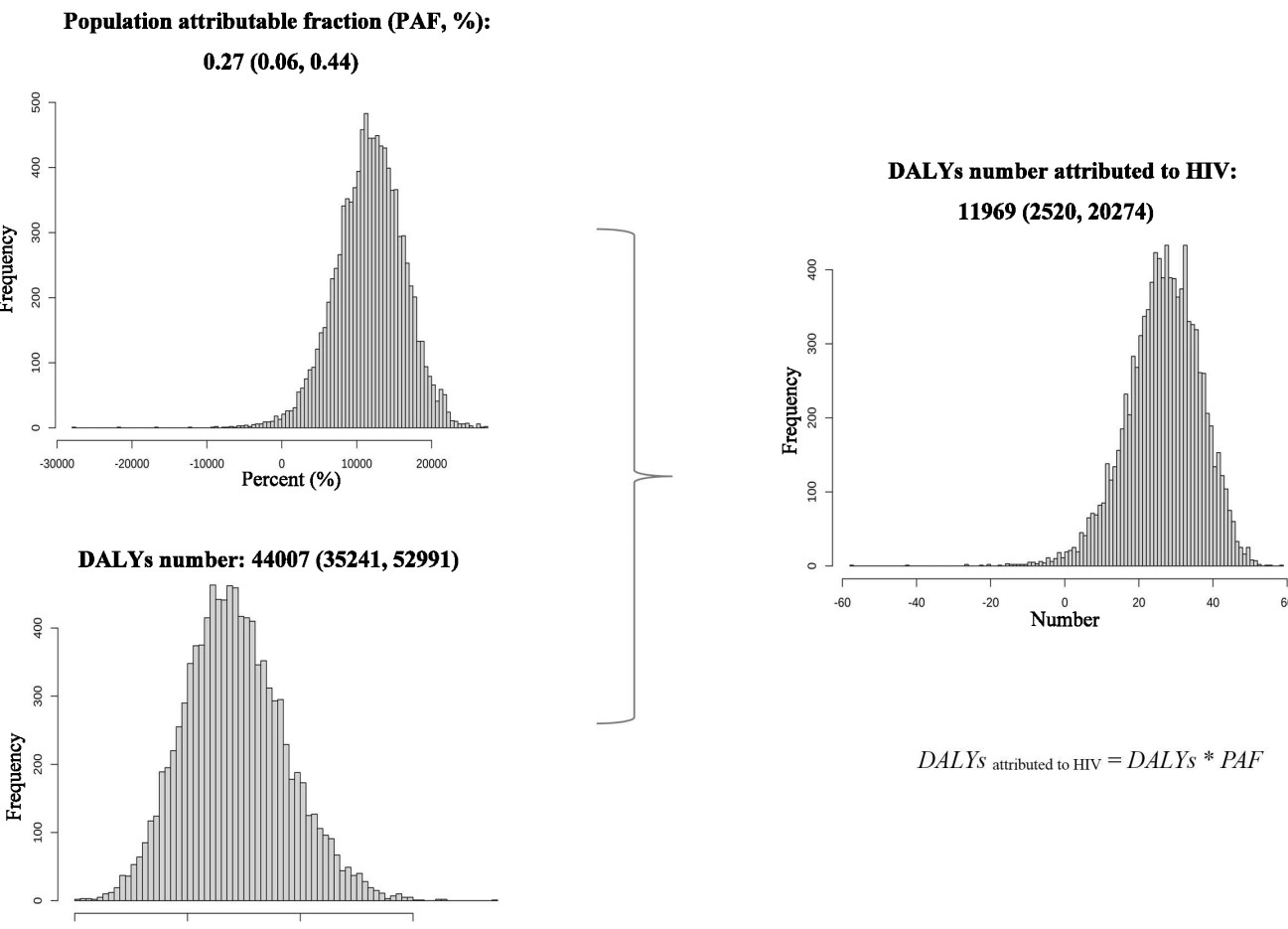

**South Africa in 2000:**

**Population attributable fraction (PAF, %):**

**0.27 (0.06, 0.44)**

**DALYs number: 44007 (35241, 52991)**

**DALYs number attributed to HIV:**

**11969 (2520, 20274)**

$$DALYs_{\text{attributed to HIV}} = DALYs * PAF$$

**Figure 2** Example procedure for calculating the HIV-associated NHL DALY in South Africa. DALY, disability-adjusted life year; NHL, non-Hodgkin's lymphoma; PAF, population attributable fraction.

Exchange query tool (https://vizhub.healthdata.org/gbd-results/) in 5-year-old intervals. We will download national HIV prevalence data (≥15 years old) from 1990 to 2019 from the Joint United Nations Programme on HIV/AIDS (UNAIDS) (https://aidsinfo.unaids.org/). The criteria for dividing countries and regions will use UNAIDS classification and International Organization for Standardization codes alpha-3 (online supplemental file 2).

### PAF modelling implementation

Once we have obtained the pooled RR value based on meta-analysis, we will proceed with the PAF modelling to estimate the burden of HIV-associated NHL (including HIV-related DALY and age-standardised DALY rate) according to the following steps.

### Step 1: PAF (%) calculation

We will calculate PAF combined with HIV prevalence data and pooled RR, with the following assumptions: (1) the pooled RR is applicable across the global, regions and all countries; (2) we ignore various confounding biases between primary studies, which may cause slight deviation in our estimation; (3) we assume the pooled RR fits log normal distribution and beta distribution for the HIV prevalence, then we calculate 10 000 samples of PAF to obtain uncertainty intervals (2.5%, 97.5%) by R software. In practice, if eligible articles cover eight UNAIDS regions and have information on our planned subgroup analyses with sufficient data, we can calculate respective PAF with specific RR such as NHL subtype, study period, sex or age. We will estimate the PAF using the Levin's formula[16]:

$$PAF = \frac{HIV\ prevalence * (RR-1)}{1+HIV\ prevalence * (RR-1)}$$

Figure 1 illustrates the calculation process of PAF using South Africa as an example.

### Step 2: estimation of NHL DALY related to HIV

We calculate age-standardised rate (ASR) per 100 000 population of all-cause DALY and corresponding 95% CI by the direct method, based on the age-specific all-cause NHL DALY and world standard population reported in the GBD 2019.[17] Formulas for calculating the ASR are as follows:

$$Crude\ Rate = \frac{Case}{Population} \quad (1)$$

$$Proportion = \frac{Age\ Group}{\sum Age\ Group} \quad (2)$$

For each age group:

$$Adjust\ Rate = Crude\ Rate * Proportion \quad (3)$$

For aged ≥15 years old:

$$Adjust\ Rate = 100000 * sum\ (Adjust\ Rates) \quad (4)$$

Then, we assume that both number of DALY and ASR fit log normal distribution and we calculate 10 000 samples of HIV-associated DALY and ASR to obtain uncertainty interval (2.5%, 97.5%) by R software. We estimate the HIV-associated DALY number and ASR using formulas as follows:

$$DALY_{attributed\ to\ HIV} = All\_cause\ DALY * PAF \quad (1)$$

$$DALY\ ASR_{attributed\ to\ HIV} = All\_cause\ DALY\ ASR * PAF \quad (2)$$

Figure 2 shows the calculation process of the HIV-associated NHL DALY for South Africa.

### Patient and public involvement

No patient and public involved.

### DISCUSSION

The incidence rate of NHL exhibited a dramatic increase starting around 1970, and despite experiencing a plateau in between, it still recorded over 500 thousand of new cases in 2020.[4 18] The complex aetiology of NHL, coupled with the accelerated ageing process of global population, is expected to further contributed to an increase of patients and deaths number for NHL.[6 18] According to the UNAIDS, there are approximately 4000 new HIV infections reported daily, with 1100 of those are young people (aged from 15 to 24 years old).[19] Given the close relationship between HIV and NHL, it is crucial to be concerned about the trajectory of the burden of HIV-associated NHL.

Our study aims to investigate the risk of NHL in individuals with HIV infection compared with those without, and to quantify the contribution of HIV infection to incidence of NHL. To ensure the readability of the literature, we will only include articles published in English, which may limit the generalisability of our findings. In addition, it is also a limitation that we assume the pooled RR is applicable across global and regional areas. This is possible to lead to potential bias. Regarding the dissemination of our results, we will make every effort to publish them in a peer-reviewed journal as an academic article and present our findings at relevant field meetings. Since this review is based on published data, an ethical review is not required. Considering the global impact of HIV-associated NHL, we believe that conducting research on estimating the global burden of the disease can provide valuable insights for developing targeted prevention and control strategies, thereby achieving significant benefits.

**Contributors** YC, YY, ZW, ZS and PS designed the study protocol. YC and ZS prepared the first draft. YY, ZW and YL reviewed and revised the first draft. All authors read and approved the final manuscript.

**Funding** This work was supported by Science Foundation for Young Scholars of Sichuan Provincial People's Hospital (grant number NO. 2022QN18) and Subject Founds of Health Care for Cadres of Sichuan Province (grant number NO. 2022-220).

**Competing interests** None declared.

**Patient and public involvement** Patients and/or the public were not involved in the design, or conduct, or reporting, or dissemination plans of this research.

**Patient consent for publication** Not applicable.

**Provenance and peer review** Not commissioned; externally peer reviewed.

**ORCID iD**
Yan Chen http://orcid.org/0009-0009-4834-8085

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
