## [Reviewer comments · BMJ Open]

ARTICLE DETAILS

TITLE (PROVISIONAL)	Global and regional burden estimation of HIV-associated non-Hodgkin lymphoma: a meta-analysis and modeling analysis protocol
AUTHORS	Chen, Yan; Sun, Zhaochen; Sun, Ping; Liu, Yuping; Wan, Zhengwei; Ye, Yunli

VERSION 1 – REVIEW

REVIEWER	Rapiti, Nadine University of KwaZulu-Natal
REVIEW RETURNED	10-Aug-2023

GENERAL COMMENTS	Global and regional burden estimation of HIV-associated Non-Hodgkin Lymphoma: a meta-analysis and modeling analysis protocol Good study objective, with queries listed below. Protocol does need review by a statistician. 1) Correct tense and grammar throughout 2) “with common subtypes including diffuse large B cell lymphoma (DLBCL), Burkitt lymphoma, follicular lymphoma, primary central nervous system (CNS) lymphoma, and Mantle cell lymphoma”. Suggest substituting the use of the word “common” by providing % incidence or prevalence. PCNSL and mantle cell lymphomas are not regarded as common lymphomas. 3) “Although NHL is considered an AIDS-defined disease, no meta-analysis has quantified the pooled relative risk of NHL in HIV-infected individuals compared to those without HIV infection” Only high-grade NHL are AIDS-defining lymphomas. Is the study including ALL NHL or AIDS-defining lymphoma only? 4) “We will calculate PAF combined with HIV prevalence and RR value, with the following assumptions: 1) the pooled RR is applicable across the global, regions and all countries;” In Step 1 in calculating the PAF, the assumption of the pooled RR is assumed to be applicable throughout the globe. Is this assumption a study limitation?
---

	5) Clarity required; why was the age of 15 years and upward chosen? Most studies report patients above the age of 12 or 16, or adult patients.
--	--

REVIEWER	Aronoff, Stephen Temple University, Pediatrics
REVIEW RETURNED	11-Sep-2023

GENERAL COMMENTS	This study plans to quantify the effects of HIV infection on the incidence of NHL. The authors plan to estimate the HIV associated rate by meta analysis. They also plan to estimate the population attributable fraction of disease attributable to HIV using global data. I have several comments:  1. The authors should specifically state how they will modify their meta-analysis in the face of heterogeneity. This should include a discussion of random effects modeling. 2. For the PAF analysis, the authors state that they are assuming that the pooled RR is applicable across all global and regional areas. This assumption is of concern since it implies equal access to comparable care across all regions. The authors may want to consider adjusting these rates based on local HIV mortality rates or median patient age.
--

REVIEWER	Brooks, Meredith Boston University School of Public Health, Global Health
REVIEW RETURNED	12-Sep-2023

GENERAL COMMENTS	Thank you for the opportunity to review this manuscript, entitled "Global and regional burden estimation of HIV-associated Non-Hodgkin Lymphoma: a meta-analysis and modeling analysis protocol." I have reviewed the study protocol with a particular emphasis on the statistical methods and analyses proposed by the authors. First and foremost, the search strategy seems appropriate and thorough. However, for (1) title and abstract screening and (2) full text and eligible records assessment, it is recommended that tiered exclusion criteria be employed to more accurately tally the numbers excluded for specific reasons. Regardless of this, the full detailed criteria for inclusion and exclusion need to be listed in full to allow for adequate determination of appropriateness of final included articles. The authors note that they will calculate a pooled relative risk. However, with the varying study designs and effect estimates pre-determined to be eligible for inclusion, it seems unlikely that this can be done accurately. The authors have not provided enough information about how this pooled RR is calculated from these values to determine whether methods were appropriate or not. Several assumptions for calculating the PAF are difficult to justify, primarily (1) the RR value per previous comment and (2) assuming the pooled estimate is globally applicable.
---

REVIEWER	da Silva, Pedro Henrique Rodrigues University of São Paulo
REVIEW RETURNED	14-Sep-2023

GENERAL COMMENTS	I highly recommend the publication of this study protocol
---

VERSION 1 – AUTHOR RESPONSE

Reviewer #1: Good study objective, with queries listed below. Protocol does need review by a statistician.

Response: Dear Dr. Nadine Rapiti, thank you very much for your valuable comments. The co-corresponding author, Dr Zhengwei Wan, holds a PhD in Epidemiology and statistics and this protocol was reviewed by him again after revision. Then we have revised contents according to his comments.

1) Correct tense and grammar throughout

Response: Thank you for your suggestion. We have revised and improved the tense and grammar throughout for the overall manuscript.

2) “with common subtypes including diffuse large B cell lymphoma (DLBCL), Burkitt lymphoma, follicular lymphoma, primary central nervous system (CNS) lymphoma, and Mantle cell lymphoma”.

Suggest substituting the use of the word “common” by providing % incidence or prevalence. PCNSL and mantle cell lymphomas are not regarded as common lymphomas.

Response: Thank you for your suggestion. Exactly, percentage of incidence or prevalence can describe epidemiology of subtypes more clearly than the word “common”. We have replaced the content and removed PCNSL and mantle cell lymphomas.

Introduction, page 4: “According to the World Health Organization (WHO) classification, the most common NHL in Western countries are diffuse large B cell lymphoma (DLBCL), accounting for around 31% of adult cases, follicular lymphoma (FL) (22%) and Burkitt lymphoma (BL) (2%).(Swerdlow et al. 2016) (Thandra et al. 2021)”

3) “Although NHL is considered an AIDS-defined disease, no meta-analysis has quantified the pooled relative risk of NHL in HIV-infected individuals compared to those without HIV infection”

Only high-grade NHL are AIDS-defining lymphomas. Is the study including ALL NHL or AIDS-defining lymphoma only?

Response: Thank you for your comment. Indeed, we intend to include all NHL in study selection phase and do not distinguish specific subtype for the calculation of pooled RR. However, in response to the point you made, we could add another subgroup analysis for different NHL subtypes if we are able to extract enough studies with subtype information. Even we can further calculate specific RR for NHL subtype to improve our analyses.

Method, page 9: “If we can obtain sufficient data, subgroup analyses will be utilized to find the potential source of heterogeneity, including NHL subtypes, study period, population age, HIV mortality rate, study conducted region, study design types, setting of study (multi-center or single center), article quality risk level.”

Method, page 10: “In practice, if eligible articles cover eight UNAIDS regions and have information on our planned subgroup analyses with sufficient data, we can calculate respective specific PAF such as NHL subtype, period, or age-specific RR.”

4) “We will calculate PAF combined with HIV prevalence and RR value, with the following assumptions: 1) the pooled RR is applicable across the global, regions and all countries;”

In Step 1 in calculating the PAF, the assumption of the pooled RR is assumed to be applicable throughout the globe. Is this assumption a study limitation?

Response: Thank you for your comments. Indeed, it is a limitation that the pooled RR is assumed to be applicable throughout the globe. Countries are divided by the Joint United Nations Programme on HIV/AIDS (UNAIDS) into eight regions and calculating PAF based on specific-region RR is better. However, it is difficult to include studies covering every region. Existing areas may also not have enough studies (e.g., <3) to calculate a specific RR. If eligible articles cover eight UNAIDS regions with sufficient data, we can calculate respective PAF using specific-region RR. We have added it as a limitation in discussion.

Discussion, page 11: "In addition, it is also a limitation that we assume that the pooled RR is applicable across global and regional areas. This is possible to lead to potential bias."

5) Clarity required; why was the age of 15 years and upward chosen? Most studies report patients above the age of 12 or 16, or adult patients.

Response: We sincerely apologize for any confusion caused. Firstly, HIV prevalence data used to calculate population attributable fraction (PAF) is downloaded from the Joint United Nations Programme on HIV/AIDS (UNAIDS). Secondly, the description of adult in UNAIDS official reports is 15 years or older and the children's is 0-14 years.(UNAIDS 2023) In order to be consistent for population, we downloaded the number of age-specific disability-adjusted life year (DALY) for 15 years-old and above from the Global Health Data Exchange (GHDx) query tool (<https://vizhub.healthdata.org/gbd-results/>) in 5 years-old intervals. Moreover, we will also exclude the studies which include below 15 years-old people in study selection. Therefore, we chose the age of 15 years and upward as the study population.

Reviewer 2: This study plans to quantify the effects of HIV infection on the incidence of NHL. The authors plan to estimate the HIV associated rate by meta-analysis. They also plan to estimate the population attributable fraction of disease attributable to HIV using global data. I have several comments.

Response: Dear Dr. Stephen Aronoff, thank you for your suggestion. We have provided the discussion of random effects modeling and supplemental information about PAF calculation based on your suggestions as following:

1.The authors should specifically state how they will modify their meta-analysis in the face of heterogeneity. This should include a discussion of random effects modeling.

Response: Thank you for your suggestion. Firstly, we will use a chi-squared test on Cochrane's Q statistic to evaluate the heterogeneity and the Higgins's I² value to quantify the heterogeneity. Secondly, we will undertake subgroup analyses to find potential source of heterogeneity, including study conducting region, study design type, setting of study, risk level of article quality. Thirdly, when the I² value exceeds 50% we will choose random effects model and fixed effects model will be used when I² value is less than or equal to 50%. Moreover, a meta-regression will be performed to ensure whether these factors can explain the heterogeneity and the degree to which they account for it. Method, page 9: "the heterogeneity will be assessed using the Chi-squared test on Cochrane's Q statistic, and the quantitation of heterogeneity will be measured by the Higgins's I² value presented as a percentage.(Higgins and Thompson 2002) When the I² value exceeds 50% we will choose random effects model and fixed effects model will be used when I² value is less than or equal to 50%."

2. For the PAF analysis, the authors state that they are assuming that the pooled RR is applicable across all global and regional areas. This assumption is of concern since it implies equal access to comparable care across all regions. The authors may want to consider adjusting these rates based on local HIV mortality rates or median patient age.

Response: Thank you for your comments. Indeed, assumption that the pooled RR is applicable across all global and regional areas is not enough prudent. We cannot ensure that every UNAIDS region will include enough studies. However, if eligible articles cover eight UNAIDS regions with sufficient data, we will calculate respective PAF using specific-region RR to reduce potential bias.

Moreover, if we have appropriate and enough data, we will calculate HIV mortality rate or age-specific RR and PAF.

Method, page 9: "If we can obtain sufficient data, subgroup analyses will be utilized to find the potential source of heterogeneity, including NHL subtypes, study period, population age, HIV mortality rate, study conducted region, study design types, setting of study (multi-center or single center), article quality risk level."

Method, page 10: "In practice, if eligible articles cover eight UNAIDS regions and have information on our planned subgroup analyses with sufficient data, we can calculate respective specific PAF such as NHL subtype, period, or age-specific RR."

Reviewer 3: Thank you for the opportunity to review this manuscript, entitled "Global and regional burden estimation of HIV-associated Non-Hodgkin Lymphoma: a meta-analysis and modeling analysis protocol." I have reviewed the study protocol with a particular emphasis on the statistical methods and analyses proposed by the authors.

Response: Dear Dr. Meredith Brooks, thank you very much for your valuable suggestions and comments on the methodological aspects of our protocol. We have carefully reviewed the statistical methods and analyses, aiming to achieve more accurate results. Below, I have provided responses to each of your suggestions and comments, and I hope they meet with your approval.

First and foremost, the search strategy seems appropriate and thorough. However, for (1) title and abstract screening and (2) full text and eligible records assessment, it is recommended that tiered exclusion criteria be employed to more accurately tally the numbers excluded for specific reasons. Regardless of this, the full detailed criteria for inclusion and exclusion need to be listed in full to allow for adequate determination of appropriateness of final included articles.

Response: Thank you for your suggestions. This protocol is about the association between HIV and NHL limited to human. For title screening, two reviewers will firstly exclude records that contain mice, mouse, review, case report, drug trial or other non-target exposure and outcome. Secondly, abstract screening needs to determine object of study and whether there is a target type of effect value. If there is any uncertainty, it is necessary to read the full text. Thirdly, we read full text throughout to reconfirm and extract data. Cohort study, case-control study, or cross-sectional study with effective control group is eligible study design. Last, different exclusion reasons and number of records will be listed in flow chat one by one (e.g., non-target study population, no needed effect value, and population overlapping). We have described the full detailed criteria for inclusion and exclusion in revised manuscript.

Method, page 6: "For inclusion criteria, our target design types are cohort studies, case-control studies or cross-sectional studies with effective control group. These studies need to expound the association between HIV and NHL using target effect value. In addition, eligible studies should provide detailed participant information, including the total number of participants, the number of participants in case/exposure and control/non-exposure groups, as well as the quantity of NHL cases in case/exposure group. For exclusion criteria, we will exclude records whose type are review, opinion pieces, poster presentations, letters, case reports or case series. If there is no effective effect value about association between HIV and NHL or it is not target study population, this record will be also excluded. In addition, when there is population overlap in multiple articles, we will choose the one with larger study size and better article quality. As regards duplicate records, we will exclude incomplete or earlier period version."

The authors note that they will calculate a pooled relative risk. However, with the varying study designs and effect estimates pre-determined to be eligible for inclusion, it seems unlikely that this can be done accurately. The authors have not provided enough information about how this pooled RR is calculated from these values to determine whether methods were appropriate or not.

Response: We sincerely apologize for any inconvenience caused. As you pointed out, study designs and effect estimates pre-determined should be chosen prudently to ensure accuracy of estimation.

This study on the association between HIV and NHL does not involve intervention experiments, so we chose observational cohort studies, case-control studies, or cross-sectional studies with effective control group. In the follow-up implementation, we will use RR, OR and SIR to calculate the pooled RR. As regards SIR, according to previous literature, when disease incidence is rare, SIR can approximate RR. (Stelzle et al. 2021) Because NHL is relatively rare, SIR is also an appropriate choice.

After completing study selections to obtain effect value and corresponding 95% confidence interval, meta-analyses will be performed using R software (version 4.2.2) through the package 'meta'. For the pooled RR, we will use function 'metagen' to pool results of multiple independent studies. In this process, we will choose appropriate model to calculate pooled RR. When the I² value (quantification of heterogeneity) exceeds 50% we will choose random effects model and fixed effects model will be used when I² value is less than or equal to 50%. This information has been included in our revised manuscript. (Discussion, Page 8)

Method, page 8: "Firstly, we will calculate a pooled relative risk (RR) and the corresponding 95% confidence intervals (95% CI) using function 'metagen' included in package 'meta' to quantitatively estimate the association between HIV and NHL. Then the heterogeneity will be assessed using the Chi-squared test on Cochrane's Q statistic, and the quantitation of heterogeneity will be measured by the Higgins's I² value presented as a percentage. (Higgins and Thompson 2002) When the I² value exceeds 50% we will choose random effects model and fixed effects model will be used when I² value is less than or equal to 50%."

Several assumptions for calculating the PAF are difficult to justify, primarily (1) the RR value per previous comment and (2) assuming the pooled estimate is globally applicable.

Response: Thank you very much for your comments. For the RR value, as we described above that it is a limitation that the pooled RR is assumed to be applicable throughout the globe. Meanwhile, we can not ensure that every UNAIDS region will include enough studies. However, if eligible articles cover eight UNAIDS regions with sufficient data, we will calculate respective PAF using specific-region RR to reduce potential bias.

Discussion, page 11: "In addition, it is also a limitation that we assume that the pooled RR is applicable across global and regional areas. This is possible to lead to potential bias."

Method, page 9: "If we can obtain sufficient data, subgroup analyses will be utilized to find the potential source of heterogeneity, including NHL subtypes, study period, population age, HIV mortality rate, study conducted region, study design types, setting of study (multi-center or single center), article quality risk level."

Method, page 10: "In practice, if eligible articles cover eight UNAIDS regions and have information on our planned subgroup analyses with sufficient data, we can calculate respective specific PAF such as NHL subtype, period, or age-specific RR."

Reviewer 4: I highly recommend the publication of this study protocol.

Response: Dear Dr. Pedro Henrique Rodrigues da Silva, thank you very much for your recognition of our protocol. We actively replied to and revised the comments from other reviewers, hoping to publish it successfully.

Reference

Higgins JP, Thompson SG. 2002. Quantifying heterogeneity in a meta-analysis. *Stat Med*. Jun 15;21:1539-1558. Epub 2002/07/12.

Stelzle D, Tanaka LF, Lee KK, Ibrahim Khalil A, Baussano I, Shah ASV, McAllister DA, Gottlieb SL, Klug SJ, Winkler AS, et al. 2021. Estimates of the global burden of cervical cancer associated with HIV. *Lancet Glob Health*. Feb;9:e161-e169. Epub 2020/11/20.

Swerdlow SH, Campo E, Pileri SA, Harris NL, Stein H, Siebert R, Advani R, Ghielmini M, Salles GA, Zelenetz AD, et al. 2016. The 2016 revision of the World Health Organization classification of lymphoid neoplasms. *BLOOD*. May 19;127:2375-2390. Epub 2016/03/17.

Thandra KC, Barsouk A, Saginala K, Padala SA, Barsouk A, Rawla P. 2021. Epidemiology of Non-Hodgkin's Lymphoma. Med Sci (Basel). Jan 30;9. Epub 2021/02/13.
Fact sheet - Latest global and regional statistics on the status of the AIDS epidemic.: Joint United Nations Programme on HIV/AIDS (UNAIDS)

VERSION 2 – REVIEW

REVIEWER	Rapiti, Nadine University of KwaZulu-Natal
REVIEW RETURNED	14-Nov-2023
GENERAL COMMENTS	“Ethics and dissemination Our study is based on published articles thus the ethnic approval is not essential.” Correct spelling of ethic Other corrections are accepted.

VERSION 2 – AUTHOR RESPONSE

Reviewer #1: “Ethics and dissemination Our study is based on published articles thus the ethnic approval is not essential.” Correct spelling of ethic. Other corrections are accepted.

Response: Dear Dr. Nadine Rapiti , thank you very much for your comments. We are glad that other corrections are accepted. And we have corrected the spelling of “ethic” (Page 2 “Ethics and dissemination”).